# Evaluation of Polyphenol Content and Antioxidant Capacity of Aqueous Extracts from Eight Medicinal Plants from Reunion Island: Protection against Oxidative Stress in Red Blood Cells and Preadipocytes

**DOI:** 10.3390/antiox9100959

**Published:** 2020-10-07

**Authors:** Eloïse Checkouri, Franck Reignier, Christine Robert-Da Silva, Olivier Meilhac

**Affiliations:** 1INSERM, UMR 1188 Diabète athérothombose Réunion Océan Indien (DéTROI), Université de La Réunion, 97490 Sainte-Clotilde, La Réunion, France; eloise.checkouri@gmail.com (E.C.); christine.robert@univ-reunion.fr (C.R.-D.S.); 2Habemus Papam, Food Industry, 97470 Saint-Benoit, La Réunion, France; dg.asbethleem@outlook.fr; 3CHU de La Réunion, CIC 1410, 97410 Saint-Pierre, La Réunion, France

**Keywords:** antioxidant, medicinal plants, polyphenols, infusions, decoctions, pharmacopeia

## Abstract

Background—Medicinal plants are traditionally used as infusions or decoctions for their antioxidant, anti-inflammatory, hypolipidemic and anti-diabetic properties. Purpose—The aim of the study was to define the polyphenol composition and to assess the antioxidant capacity of eight medicinal plants from Reunion Island referred to in the French Pharmacopeia, namely *Aphloia theiformis*, *Ayapana triplinervis*, *Dodonaea viscosa*, *Hubertia ambavilla*, *Hypericum lanceolatum*, *Pelargonium x graveolens*, *Psiloxylon mauritianum* and *Syzygium cumini*. Methods—Polyphenol content was assessed by biochemical assay and liquid chromatography coupled to mass spectrometry. Antioxidant capacity was assessed by measuring DPPH reduction and studying the protective effects of herbal preparation on red blood cells or preadipocytes exposed to oxidative stress. Results—Polyphenol content ranged from 25 to 143 mg gallic acid equivalent (GAE)/L for infusions and 35 to 205 mg GAE/L for decoctions. Liquid chromatography coupled to mass spectrometry analysis showed the presence of major bioactive polyphenols, such as quercetin, chlorogenic acid, procyanidin and mangiferin. Antioxidant capacity assessed by different tests, including DPPH and Human red blood cell (RBC) hemolysis of herbal preparations, demonstrated a dose-dependent effect whatever the extraction procedure. Our data suggest that decoction slightly improved polyphenol extraction as well as antioxidant capacity relative to the infusion mode of extraction (DPPH test). However, infusions displayed a better protective effect against oxidative stress-induced RBC hemolysis. Conclusion—Traditional preparations of medicinal plant aqueous extracts (infusions and decoctions) display antioxidant properties that limit oxidative stress in preadipocytes and red blood cells, supporting their use in the context of metabolic disease prevention and treatment.

## 1. Introduction

Reunion Island is a hotspot of vegetal biodiversity where the use of medicinal plant-derived beverages is documented since the 17th century [1]. To date, 22 plants are referred to in the French Pharmacopeia; they are traditionally used by the population for their antioxidant, antidiabetic, hypolipidemic and anti-inflammatory properties. The commercialization of these plants (dry form) for infusion or decoction is based on empirical data, urging for the need to investigate the composition and antioxidant activity of these herbal preparations.

Eight commercially available medicinal plants were selected for this study according to the following criteria: (1) wide use in traditional practices in Reunion Island, (2) reported biological activity: anti-diabetic, antioxidant and cholesterol-lowering properties, and (3) marketability. The bioactivity and traditional uses of the different medicinal plants is described in the Appendix A. 

Several ethnopharmacological studies have identified bioactive molecules from medicinal plants exerting antioxidant and anti-inflammatory activity. These studies put in light the potential of the use of medicinal plants in the prevention and management of metabolic disorders in which excessive oxidative stress plays an important role.

Oxidative stress is the consequence of an imbalance between the production of free radicals and antioxidant defense [2]. An excess of reactive oxygen species (ROS, such as ^•^OH or O_2_^•^^−^) can induce major damage on the main cell constituents including lipids, proteins, carbohydrates and nucleic acids. Oxidative stress is known to be a trigger for several signaling pathways, leading to chronic, low-grade inflammation (production of cytokines, activation of different cell types by increasing the expression of adhesion molecules for leukocytes, etc.) and thus participating in the etiology of several human diseases such as diabetes mellitus and obesity [3,4,5,6]. In the context of metabolic diseases, oxidative stress and inflammation are the first signs of cellular dysfunction prior to the onset of symptoms. For example, in case of intestinal dysbiosis and unbalanced diet, preadipocytes and adipocytes are subjected to a high metabolic activity. In fact, the excess of energy provided by the diet leads to an increased stimulation of the energetic metabolism (mitochondrial activity and respiratory chain) causing overwhelming production of free radicals resulting in a marked oxidative stress. Oxidative stress can impair preadipocyte differentiation and promotes insulin resistance [7,8].

Several studies have focused on dietary polyphenols [9]. Medicinal plants constitute natural sources of polyphenols and are commonly used in traditional medicine all over the world. Polyphenols are secondary plant metabolites with several hydroxyl groups on one or more aromatic rings. They are omnipresent in the plant kingdom, notably in vascular plants, encompassing more than 8000 identified components, including simple molecules but also high-molecular weight polymers. Polyphenols are the most abundant dietary antioxidants as their dietary intake is about 1 g/day [10,11,12,13].

Evidence for the role of these abundant dietary micronutrients in the prevention of degenerative diseases such as Alzheimer’s disease or cardiovascular disease is emerging [14,15,16,17]. For example, various bioactive effects of flavonoids have been investigated, highlighting their anti-inflammatory, antioxidant, anti-diabetic, cardioprotective and anti-cancer properties via their action on different enzymes and signaling pathways [18,19,20,21,22,23]. Moreover, several studies have shown that dietary polyphenols exert antioxidant activity and display protective effects on erythrocyte oxidative damage and lipid peroxidation [24,25,26,27,28]. Among the plants selected for this study, some have been described in the literature while others remain poorly documented, in particular because they are rare and endemic. Despite their widespread use in Reunion Island, there is a real lack of information on the chemical composition and biological properties of herbal teas (infusion or decoction) prepared from these medicinal plants. The goal of this study was to explore the polyphenol composition (qualitative and quantitative) of eight different Reunion Island medicinal plants and to assess their antioxidant capacity according to their preparation mode (polyphenol rich extract, infusion or decoction) by different approaches in vitro and in cell culture. 

## 2. Material and Methods

### 2.1. Sample Preparation

#### 2.1.1. Raw Material

The following industrial process was applied for the different plant leaves and stems (except for *S. cumini* for which dehydrated seed powder was used): trimming, microwave dehydration, grinding and packaging in tea bags containing 1 g of dehydrated plant mash or powder. All raw material was provided by HABEMUS PAPAM Industry. Industrial lots and corresponding GPS coordinates are provided in Table 1.

#### 2.1.2. Polyphenol-Rich Extracts (PRE)

Plant powders were incubated with an acetone/water (70/30) solvent mixture at 4 °C for 90 min (4 g of plant mash in 20 mL acetone/water). The mixture was then centrifuged at 1372 g, 4 °C for 20 min. The polyphenol-rich supernatant (ranging from 7 to 15 mL depending on the level of solvent retention by the plant material) was collected and stored at −80 °C until analysis. Each sample was prepared in triplicate.

#### 2.1.3. Infusions and Decoctions

Infusions were prepared by the addition of one tea bag (containing 1 g of plant mash) to 1 L of boiling distilled water followed by 15 min of incubation at room temperature (RT). For decoction, one tea bag was added to 1 L of RT distilled water and kept boiling for 30 min. The tea bags were removed and the infusions or decoctions were cooled at RT before storage at −80 °C until analysis.

## 3. Determination of Antioxidant Polyphenol Content in Medicinal Plant Extracts, Infusions and Decoctions

### 3.1. Total Phenolic Content

To determine the total polyphenol content of the different samples, the Folin–Ciocalteu test was used [29]. Briefly, 25 µL of each sample, 125 µL of Folin–Ciocalteu’s phenol reagent (Sigma-Aldrich, St-Louis, MO, USA) and 100 µL of sodium carbonate (75 g/L) were added to a 96-well microplate and incubated at 50 °C for 5 min. After cooling for 5 min at 4 °C, the absorbance was measured at 760 nm (Fluostar Omega, Bmg Labtech, Cambridge, UK). A calibration curve was made using a standard solution of gallic acid (Sigma-Aldrich, St-Louis, MO, USA). For the polyphenol-rich extracts (PRE), the total polyphenol content was expressed as g gallic acid equivalent (GAE)/1 g plant mash. For infusions and decoction, the total polyphenol content was expressed as mg GAE/1 g (=mg GAE/1 L since these preparations are made at a ratio of 1 g/L)

### 3.2. Determination of Antioxidant Polyphenol Content in Medicinal Plant PRE, Infusions and Decoctions

Polyphenols contained in medicinal plant PRE, infusions and decoctions were identified by ultra-high-performance liquid chromatography analysis coupled to diode array detection and electrospray ionization-mass spectrometry (UPLC-ESI-MS, Agilent Technologies, Les Ulis, France) in Polyphenols Biotech platform (ADERA, Villenave d’Ornon, France). Briefly, samples were diluted, filtered on PTFE (0.45 µm) and injected at 1 µL on an Agilent ZORBAX RRHD SB-C18 column (1.8 µm, 2.1 × 100 mm) for chromatographic separation. The content of the column was eluted with a gradient mixture of 0.1% formic acid in water (A) and 0.1% formic acid in acetonitrile (B) at the flow rate of 0.3 mL/min, with 1% B at 0–0.4 min, 1–10% B at 0.4–2 min, 10–35% B at 2–6 min, 35–50% B at 6–7 min, 50–70% B at 7–8.8 min, 70–92% B at 8.8–10.8 min, 92–100% B at 10.8–12 min, 100–1% B at 12–15.2 min. The column temperature was held at room temperature and the detection wavelength was set to 280, 320 and 380 nm. For the mass spectrometer conditions, an electrospray ionization source was used. 

## 4. Evaluation Antioxidant Activity of Polyphenol-Rich Extracts, Infusions and Decoctions from Medicinal Plants

### 4.1. Evaluation of the Free Radical-Scavenging Activity: DPPH Assay

Evaluation of the different sample radical scavenging activity on DPPH was performed according to the method described by Yang et al. with slight modifications [30]. Briefly, 0.25 mM of DPPH (Sigma-Aldrich, St-Louis, MO, USA) was diluted in methanol and added to a 96-well microplate containing plant PRE, infusion or decoction samples. An antioxidant standard of 100 µM vitamin C (Sigma-Aldrich, St-Louis, MO, USA) was used as a positive control. After 25 min of incubation at 25°C, the absorbance was measured at 517 nm (Fluostar Omega, Bmg Labtech, Cambridge, UK). The percentage of free radical scavenging activity was determined based on the optical density (OD) measured at 517 nm with a spectrophotometer, according to the following formula:
Antioxidant capacity (%) = [(OD negative control − OD sample)/(OD negative control)] × 100

The negative control is DPPH solution supplemented in sample vehicle diluted in methanol following sample dilution conditions.

### 4.2. Evaluation of the Protective Effects on Erythrocyte Damage Induced by the Radical AAPH: Hemolysis Test

The capacity of the plant polyphenol-rich extracts (PRE), infusions and decoctions to inhibit free radical-induced hemolysis was measured as described [26]. Red blood cells were obtained from five healthy subjects (O + group) at the “French Blood Agency”. Cells were suspended in NaCl (0.15 M) at pH 7; different concentrations of PRE (1, 5, 10, 25 and 50 µM GAE) were added and hemolysis was triggered by addition of AAPH (50 mM). Red blood cell hemolysis was followed at 450 nm for 18.5 h at 37 °C using a FLUOstar Optima spectrophotometer (Bmg Labtech). Half time hemolysis (HT50) was determined with the x-axis value corresponding to ((OD t_0_ + OD endpoint)/2) value of ordinate axis.

### 4.3. Cell Culture 

Mouse 3T3-L1 preadipocytes were obtained from the American Type Culture Collection (ATCC©, Manassas, VA, USA). Cells were grown in Dulbecco’s modified Eagle’s medium containing 25 mM glucose, 10% heat-inactivated fetal bovine serum, 5 mM L-glutamine, 2 µg/mL streptomycin and 0.03 µg/mL penicillin (Pan Biotech, Dutscher, Brumath, France). Cells were maintained in a humidified 5% CO_2_ incubator at 37 °C.

#### 4.3.1. Evaluation of the Medicinal Plant Extracts, Infusions and Decoctions on Cell Viability

##### LDH Assay

3T3-L1 preadipocytes were seeded at a density of 10 × 10^3^ cells/well for 24 h. Then, cells were incubated with PRE, infusions, decoctions or 1% Triton X-100 (Sigma-Aldrich, St-Louis, MO, USA) for 24 h. After treatment, the supernatant was collected and LDH assay was performed according to the supplier’s instructions (Abcam, Cambridge, UK).

##### Neutral Red Assay

Cell viability was assessed by measuring the intake of the neutral red as described by Repetto and colleagues with slight modifications [31]. This test is based on the capacity of viable cell lysosomes to incorporate neutral red dye. After 24 h of incubation with medicinal plant PRE, infusions and decoctions, the medium was removed and cells were washed twice with PBS. A Total of 100 µL of neutral red medium solution (40 µg/mL) was added and the plate was incubated for 2 h at RT. The medium was removed and the cells were washed twice with PBS. Finally, neutral red destain solution (49.5% water, 49.5% ethanol, 1% glacial acetic acid) was added and absorbance was read at 540 nm. Results are expressed as % of control.

#### 4.3.2. Evaluation of the Effects of Medicinal Plants PRE, Infusions and Decoctions on Intracellular ROS Production by 3T3-L1 Preadipocytes Exposed to H_2_O_2_

Intracellular levels of reactive oxygen species were determined by measuring the oxidation of 2′,7′-dichlorofluorescein diacetate (DCFDA) (Sigma-Aldrich, St-Louis, MO, USA), as previously described [29]. In a 96-well black plate, cells were seeded at a density of 10 × 10^3^ cells/well and incubated with cell culture medium containing the different medicinal plant preparations. After 24 h, cells were washed with PBS and exposed to DCFH-DA (10 µM) for 45 min. After the incubation with the probe, cells were washed and then incubated for 1 h with or without H_2_O_2_ (200 µM). ROS were detected by fluorimetry (Fluostar Omega, Bmg Labtech, Cambridge, UK) at 492 nm for excitation and 520 nm for emission.

## 5. Statistical Analysis

Data are expressed as means ± SEM of three independent experiments performed in triplicate. Statistical analysis was carried out using GraphPad Prism software (San Diego, CA, USA). Differences between the means were determined by the Bonferroni test and were considered as statistically significant for a *p* value < 0.05.

## 6. Results and Discussion

This study aimed at evaluating the composition and antioxidant properties of polyphenol-rich extracts (PRE), infusions and decoctions of eight medicinal plants from Reunion Island: *Aphloia theiformis*, *Ayapana triplinervis*, *Dodonaea viscosa*, *Hubertia ambavilla*, *Hypericum lanceolatum*, *Pelargonium x graveolens*, *Psiloxylon mauritianum* and *Syzygium cumini*.

### 6.1. Determination of Polyphenol Content in Medicinal Plant Extracts

#### 6.1.1. Total Phenolic Contents

As reported by Nowicka et al., the total polyphenol content can be used as an indicator of the antioxidant capacity of the food matrix. In our study, the total polyphenol content of the different medicinal plants was assessed by using the Folin–Ciocalteu assay and expressed in g of gallic acid equivalent (GAE)/g of plant (=GAE/L for infusions or decoctions since they were prepared using a ratio of 1 g/L) [32].

As shown on Figure 1, polyphenol contents significantly varied depending on the medicinal plant considered. *S. cumini, H. lanceolatum* and *P. mauritianum* extracts exhibited higher concentrations of polyphenols with, respectively, 7.6%, 5.2% and 4.9% GAE, *w/w*. Furthermore, these values are comparable to that of other dietary sources of polyphenols such as star anise, cocoa, spearmint, rosemary or thyme [33]. Traditional extraction consisting in infusion and decoction provided similar results, suggesting that these three aforementioned plants contained more polyphenols than the others. As expected, the decoction procedure allowed a better extraction than infusion (statistical significance was reached for *A. theiformis, S. cumini, H. lanceolatum* and *P. mauritianum*) (Figure 1B). Acetone/water extraction led to a lower yield than aqueous extraction (decoction/infusion) probably due to a different ratio of mash to solvent (4 g/20 mL acetone-water then reduced to about 10 mL of extract for PRE versus 1 g/1000 mL for decoction and infusion). Traditional preparation is therefore very efficient for extracting polyphenols.

#### 6.1.2. Polyphenol Composition of Medicinal Plant PRE, Infusions and Decoctions by UPLC-MS/MS Analysis

Ultra-performance liquid chromatography- tandem mass spectrometry (UPLC-MS/MS) analysis performed on both infusion and decoction samples of *Aphloia theiformis*, *Ayapana triplinervis*, *Dodonaea viscosa*, *Hubertia ambavilla*, *Hypericum lanceolatum*, *Pelargonium x graveolens*, *Psiloxylon mauritianum* and *Syzygium cumini* allowed identification of 2 major families of polyphenols: phenolic acids and flavonoids (Table 2).

Polyphenol composition was different depending on the plant considered. *A. theiformis* contained almost exclusively mangiferin and its derivatives as reported by Dantu et al. [34]. This polyphenol, widely contained in mangos, is known to exert antioxidant activity and to inhibit both glucose uptake and carbohydrate metabolism enzymes such as α-glucosidase and β-glucosidase [35,36]. The presence of this polyphenol supports the use of *A. theiformis* for its antidiabetic properties in traditional medicine. *A. triplinervis* contains a majority of quercetin glycosides but also specific polyphenols such as ayapin, ayapanin and thymohydroquinone dimethyl ether, as previously described in the literature [37]. High concentrations of catechin, procyanidin, quercetin, isorhamnetin and coumaroyl quinic acid were found in *D. viscosa*. These polyphenols are mostly flavonoids, recognized for their antioxidant and anti-inflammatory properties [21,38]. Chlorogenic acid, quercetin hexoses and di-caffeoyl quinic acid derivatives were detected in *H. ambavilla*. Hydroxybenzoic acids contained in *H. ambavilla* also have an antioxidant and healing potential [14]. Two major types of polyphenols were identified in *H. lanceolatum*: phenolic acids such as chlorogenic acid and di-caffeoyl quinic acid and two types of flavonoid represented by quercetin hexoses and procyanidin isomers. For *P. x graveolens*, polyphenols such as caffeoyl glucarate, myricetin, quercetin and kaempferol hexoses were identified. Similarly, most of the identified polyphenols in *D. viscosa* belong to the flavonoid family. *S. cumini* seed powder extracts exhibited the higher polyphenol content mainly represented by ellagitannins; vescalagin, castalagin, galloyl-hexoses, HHDP-hexose, galloyl tannin and ellagic acid were detected. This result highlights differences in polyphenol composition between fruit seeds, skin and pulp; indeed, Tavares et.al showed that *S. cumini* skin and pulp mainly contain anthocyanins, flavonols and flavanonols [39]. Finally, *P. mauritianum* extracts contained quercetin hexoses, kaempferol derivatives and also asiatic acid. Interestingly, corosolic acid identified by Rangasamy et al. was not detected in our experimental conditions [40]. This difference may be due to different environmental conditions (temperature, altitude, biotope).

The main biological activities reported in the literature for the different compound identified in our eight plant extracts are presented in the Appendix A.

Our study shows that the identified polyphenolic compounds are frequently found in human diet and medicinal plants [13,41] and are known for their bioactivity [10,15,42].

Interestingly, UPLC-MS analysis led to the detection of a wide range of polyphenols in infusions and decoctions that may constitute the basis of their bioactivity. Indeed, these components, such as quercetin, chlorogenic acid, gallic acid and kaempferol, that represent about 20 to 40% of the total polyphenol composition of *H. lanceolatum, P. x graveolens*, *S. cumini* and *P. mauritianum* are known to exert an antioxidant activity, for example, by limiting ROS production. As reported in detail in Appendix A, most of the compounds detected and identified by mass spectrometry have already been described in the literature for their antioxidant and/or anti-inflammatory bioactivity. Another example is the composition of *A. theiformis* extracts; it has been shown that mangiferin and its metabolites represent more than 95% of the total polyphenol content. Mangiferin may be responsible for *A. theiformis* antidiabetic and cholesterol lowering effects traditionally reported for this medicinal plant. In fact, the bioactivity observed in medicinal plants results from the synergistical or individual action of the different components identified in aqueous extracts, and particularly polyphenols. Although our study is intended to test the bioactivity of these eight medicinal plants, our results underline the importance of studying the polyphenol contents of these extracts. The mode of preparation did not significantly impact the polyphenol qualitative composition and further investigations are needed to understand the mechanisms underlying plant bioactivity. (Table 2).

## 7. Impact of Infusion and Decoction Process on the Antioxidant Polyphenolic Contents

Data presented in Figure 1B show that polyphenol concentrations ranged from 25 to 143 mg acid gallic equivalent (GAE)/L for infusions and 35–219 mg GAE/L for decoctions. Interestingly, polyphenol contents of infusions and decoctions (1 g/L) are comparable to that of a chocolate beverage with milk, soy milk or pure pomelo juice [33]. This result suggests that significant amounts of polyphenols can be extracted with water despite a low quantity of plant matrix (1 g/L). Indeed, plant infusions such as tea or medicinal plants can be prepared with more than 2 g of plant mash/L [43,44,45]. Statistical analysis shows an impact of the extraction process (infusion vs. decoction) on the total polyphenol content; indeed, for *A. theiformis, H. lanceolatum, S. cumini* and *P. mauritianum,* the decoction process yielded higher polyphenol contents. The difference of plant matrix immersion time and exposition at high temperature between the two modes of preparation can explain this result. Decoction process may allow the extraction of polyphenols from rigid tissues such as lignified tissues by promoting cell lysis.

### 7.1. Impact of Infusion and Decoction Process on Antioxidant Activity

#### 7.1.1. DPPH Assay

The free-radical scavenging activity of medicinal plant infusions, decoctions and acetone extracts was evaluated by the DPPH reduction assay. Data presented in Figure 2 show the percentage of reduced DPPH in the presence of infusions, decoctions or PRE at different polyphenol final concentrations (Figure 2A–C).

The results illustrate a dose-dependent antioxidant activity. This antioxidant effect is the result of the radical scavenging by polyphenol reducing extremities (hydroxyl groups). As described by Villaño et al., polyphenols can exert antioxidant activity by direct abstraction of phenol H-atom with the nitrogen of the DPPH or by undergoing an electron transfer to the free radical. The number of -OH groups available is also an important factor that impacts the antioxidant activity [46]. Infusions and decoctions demonstrated higher antioxidant activity than observed for vitamin C (100 µM, used as positive control). Interestingly, at a final concentration of 25 µM GAE, *S. cumini* infusions, decoctions and PRE exerted the best free radical-scavenging activity reaching, respectively, 60.5 ± 4.1%, 55 ± 2.7% and 58.3 ± 0.2% (Figure 2D). Overall, the antioxidant capacity of the different medicinal plant infusions and decoctions, respectively, vary from 27.22 to 60.5% and from 28.5 to 55%, for a final concentration of GAE fixed at 25 µM. Our results underline that the antioxidant capacity of *S. cumini* and *P. mauritianum* infusions are close to that observed for tropical fruits cultivated in Reunion Island that exerted radical-scavenging activities showed to reduce by 45 to 58% DPPH oxidation [29]. For a fixed concentration of polyphenols, a trend towards an improved antioxidant capacity was observed for decoctions vs. infusions. Statistical significance was reached for *A. theiformis* and *H. lanceolatum* at 10 µM GAE and for *P. x graveolens* and *S. cumini* at 50 µM GAE (*p* < 0.05, data not shown).

In Figure 3, results are presented after normalization to the initial 1g/L concentration of dry plant powder diluted in water for infusion and decoction, reflecting the herbal tea that a customer would ingest. Significant differences between antioxidant activity of *H. ambavilla*, *A. triplinervis*, *D. viscosa* and *P. x graveolens* infusions vs. decoctions were observed when evaluated without normalization to polyphenol levels. This result can be linked to the differences of final polyphenol concentrations between infusions and decoctions. Other plant herbal extracts reached their maximal radical-scavenging activity.

These results are in accordance with the total polyphenol concentrations reported in Figure 1, suggesting that polyphenol account for most of the antioxidant activity of infusions and decoctions. For example, *A. triplinervis* has the lowest polyphenol content and also the lowest radical scavenging activity.

#### 7.1.2. Evaluation of the Protective Effects of Medicinal Plant PRE, Infusions and Decoctions on Red Blood Cell Damage Induced by the Radical AAPH

Oxidative stress induces biomolecule damage and alters redox intracellular signaling. Several studies highlighted the protective effects of polyphenols against oxidative stress. On the one hand, protective effects of the polyphenols rely on their ability to improve cellular antioxidant response by promoting the Nrf2 pathway or by stimulating antioxidant enzymes such as glutathione peroxidase (GPx), catalase or superoxide dismutase (SOD) [47]. On the other hand, the capacity of polyphenols to act as free-radical scavengers and electron donors give them a protective effect against oxidative stress [48]. Polyphenols prevent lipid peroxidation and cell membrane damage caused by oxidative stress [49,50].

Several reports emphasize on the use of medicinal plants and polyphenols to prevent oxidative stress damage notably in red blood cells [24,51,52,53,54]. Oxidative stress affects membrane stability and fluidity [55]. To reproduce the conditions of an oxidative stress, AAPH was added to purified human red blood cells. Interactions between red blood cell membrane polyunsaturated fatty acids and AAPH-derived free radicals leads to membrane rupture and eventually hemolysis (detectable through a progressive decrease in optical density at 450 nm) [56]. In our study, the ability of plant preparations to protect red blood cells against oxidative stress was assessed by performing a hemolysis inhibition test.

Red blood cells exposed to oxidative stress in the presence of infusions, decoctions or PRE, presented a higher half-hemolysis time. Moreover, the half-hemolysis time was different according to the plant, the preparation mode and the final polyphenol concentration (Figure 4). 

It is important to note that PRE exert high protective effects at 25 µM by increasing the HT50 from 8.01 ± 0.16 h (for *H. ambavilla*) to 11.72 ± 0.12 h (for *D. viscosa*). As previously reported, polyphenols may prevent hemolysis and thus increase red blood cell half-life [24,26]. As presented in Figure 4, at a final concentration of 25 µM GAE, half-hemolysis time was prolonged from 2.9 ± 0.2 h to 6.9 ± 0.1 h for the infusions, from 2.3 ± 0.2 h to 5.8 ± 0.1 h for the decoctions and from 1.5 ± 0.2 h to 5.2 ± 0.1 h for the PRE. Our results suggest that, whatever the mode of extraction, polyphenols exert a protective effect on red blood cell damage induced by AAPH. Interestingly, infusions of *D. viscosa*, *H. lanceolatum*, *P. x graveolens* and *S. cumini* have a higher protective effect than decoctions.

### 7.2. Impact of Medicinal Plant Preparations on 3T3-L1 Preadipocytes

Preadipocytes are the base of adipose tissue expansion and function; they enable energy storage by differentiation mechanisms [57]. Protecting preadipocytes from oxidative stress is a major issue to preserve their functionality and to avoid metabolic disorders such as insulin resistance. 

#### 7.2.1. Evaluation of the Impact of Medicinal Plant Preparations on Cell Viability

As described in the literature, medicinal plant extracts may have an impact on cell viability and/or proliferation [58,59]. Before analyzing the protective effect that medicinal plants could exert on 3T3-L1 preadipocytes exposed to oxidative stress, we checked for their potential impact on cell viability and proliferation.

As shown in Figure 5, PRE, infusions and decoctions did not have any significant impact on cell necrosis at physiological (10 µM) and supraphysiological (25–50 µM) concentrations. The percentage of viable cells relative to controls assessed by neutral red assay did not show any impact of plant preparations on mortality or cell proliferation. At the doses tested (10–50 µM), our results showed that PRE, infusions and decoctions were not cytotoxic and did not induce proliferation (Figure 5, Figure 6).

#### 7.2.2. Evaluation of the Protective Effect of Medicinal Plant PRE, Infusions and Decoctions on 3T3-L1 Preadipocyte Intracellular Oxidative Stress in Response to H_2_O_2_

Oxidative stress plays a major role on cell dysfunction. As reported by Sies, high concentrations of hydrogen peroxide can lead to oxidative stress and DNA, protein and lipid damage [60]. ROS production is associated with important biomolecule and cell damage [61]. Medicinal plant ability to prevent oxidative stress may allow a control of metabolic disturbances by reducing cell damage and by limiting the onset of chronic, low-grade inflammation [4,7,61].

As shown in Figure 7, medicinal plant PRE, infusions and decoctions did not have a significant impact on 3T3-L1 preadipocyte intracellular oxidative stress in basal conditions.

However, results highlight the protective effects of medicinal plants against oxidative stress induced by H_2_O_2_. This protective effect is more pronounced at 25 µM. Unexpectedly, only a few PRE displayed a significant protective effect at this concentration despite their higher polyphenol diversity as compared to infusions and decoctions. This suggests that the bioavailability of polyphenols contained in PRE and water-based extractions is different, being more efficient for infusions and decoctions. Moreover, the comparative analysis between infusions and decoctions did not reveal any significant difference that could be due to the preparation mode; both significantly reduced the intracellular ROS production to the basal level, for all plant tested (Appendix A). It should also be noted that a protective effect has been observed for most of infusions and decoctions even at a low concentration of 10 µM: *H. ambavilla*, *D. viscosa, A. theiformis* infusions and *H. ambavilla*, *D. viscosa, A. theiformis, P. x graveolens, S. cumini* and *P. mauritianum* decoctions to totally inhibit ROS production induced by H_2_O_2_ at 10 µM GAE. These results point out the importance of the antioxidant activity of medicinal plant bioactive compounds that are known to exert peroxide inactivation leading to a protective effect on oxidative stress damages (mitochondrial stress, activation of redox signaling, cell death) [60,62].

## 8. Conclusions

Medicinal plants from Reunion Island represent an interesting nutritional approach for the prevention or treatment of metabolic disorders resulting from oxidative stress and inflammation. First, medicinal plants preparations contain a large amount of bioactive polyphenols that enable them to exert an important antioxidant activity. This has been evidenced by the ability of herbal preparations to scavenge DPPH and AAPH radicals. Second, medicinal plant preparations showed protective effects on preadipocyte intracellular oxidative stress production subjected to H_2_O_2_ stimulation.

The preparation mode did not have a significant impact on the medicinal plant protective effects except for red blood cell lysis where infusions were significantly more efficient than decoctions. One limitation of our study is that we analyzed plant extracts, which implies a mixture of several antioxidant compounds at different concentrations acting alone or in synergy. Further investigations would be needed to investigate the individual contribution of each component to the global antioxidant activity. In addition, depending on environmental conditions, industrial process (drying, milling, conditioning) and extraction method, the composition of plant herbal tea may slightly vary from one preparation method to another, but the main components and biological function should be globally similar.

These results scientifically support the potential of the traditional use of Reunion Island medicinal plant preparations (infusions or decoctions) for limiting oxidative stress, involved in metabolic disorders and associated complications.

## Figures and Tables

**Figure 1 antioxidants-09-00959-f001:**
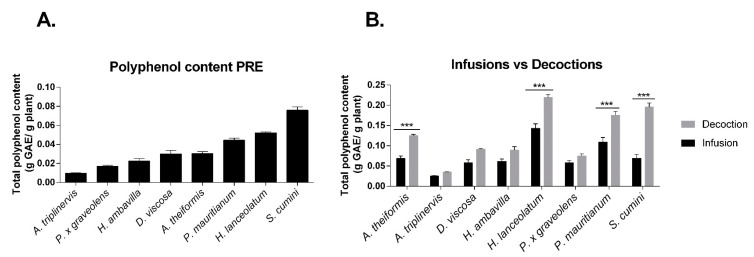
Total polyphenol content of the 8 medicinal plants depending on the preparation mode. (**A**) Medicinal plant polyphenol-rich extracts (PRE); (**B**) Infusions and decoctions. Polyphenol content was determined by Folin–Ciocalteu colorimetric assay and expressed as g gallic acid equivalent (GAE)/g plant. Infusions and decoctions were obtained with a ratio of 1 g plant/L of water, therefore g GAE/g plant = g GAE/L. Data shown are means ± SEM of three independent experiments. *** *p* < 0.005 as compared to decoctions.

**Figure 2 antioxidants-09-00959-f002:**
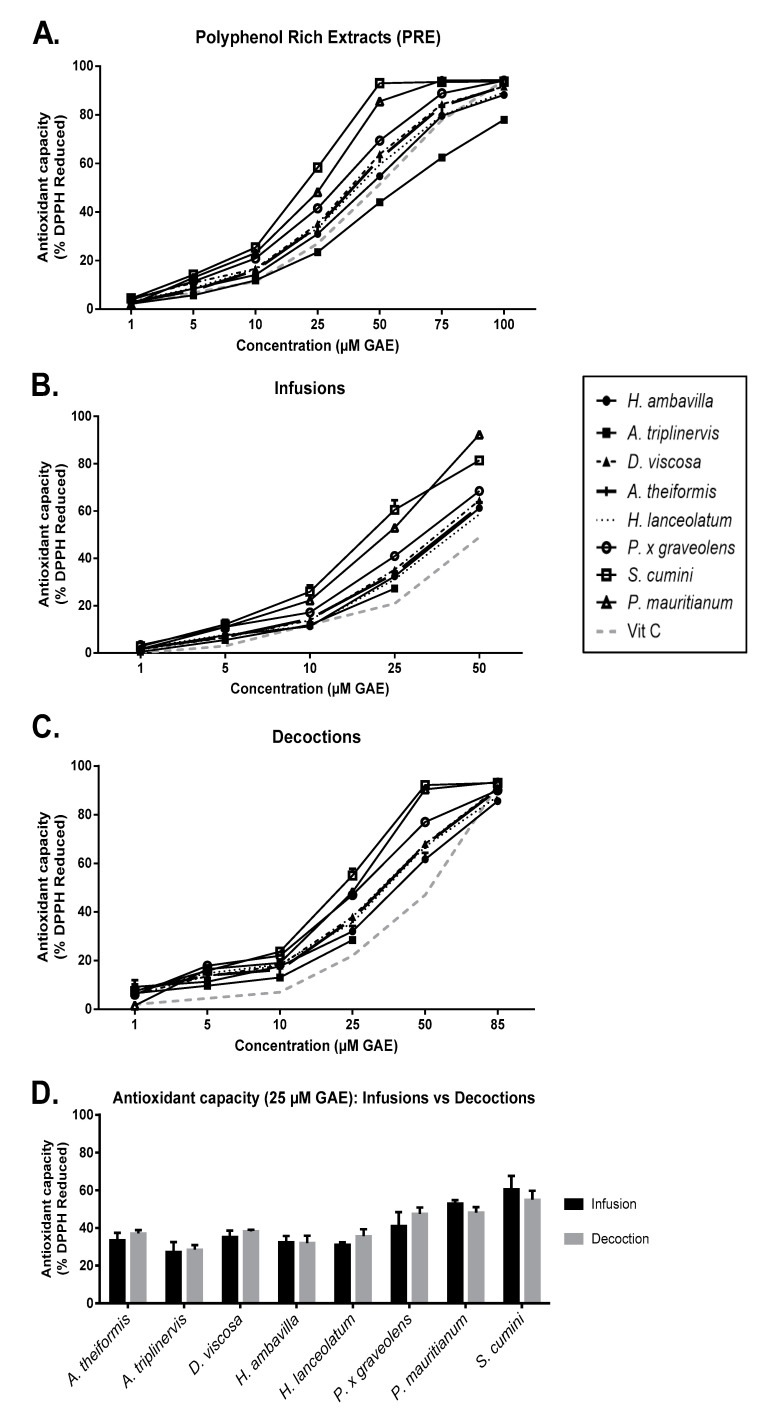
Antioxidant capacity of medicinal plant PRE, infusions and decoctions using DPPH assay after normalization to GAE. Dose-dependent antioxidant capacity of PRE (**A**), infusions (**B**) and decoctions (**C**) at different concentrations. (**D**) Antioxidant capacity of infusions versus decoctions at 25 µM GAE final concentration. The radical-scavenging activity was evaluated through the DPPH colorimetric method at 517 nm. Data are means ± SEM of three independent experiments.

**Figure 3 antioxidants-09-00959-f003:**
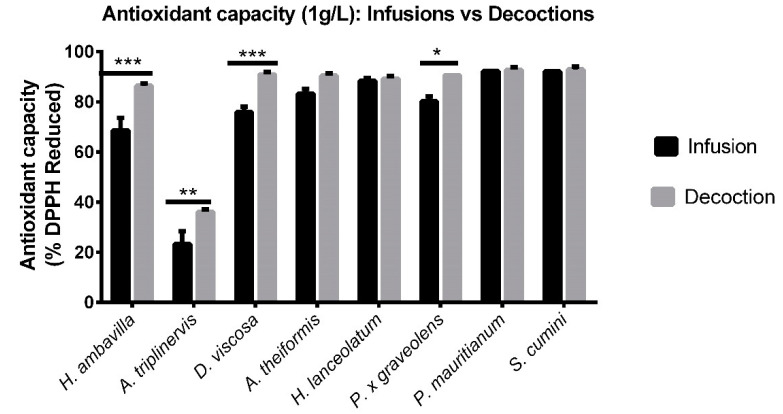
DPPH assay comparing infusion and decoction modes based on traditional preparation (1 g dry powder/L). Data are means ± SEM of three independent experiments. * *p* < 0.5, ** *p* < 0.01, *** *p* < 0.005 as compared to decoctions.

**Figure 4 antioxidants-09-00959-f004:**
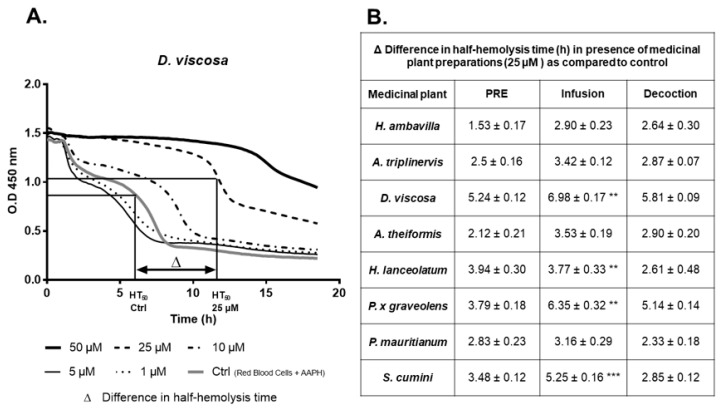
Effect of medicinal plant preparation on the half-time hemolysis (HT50) of red blood cells. Red blood cell hemolysis was induced by AAPH. Optical density at 450 nm was measured every 3 min for 18.5 h. The decrease in optical density indicated the progression of hemolysis. (**A**) Example of hemolysis kinetics when red blood cells where co-incubated with AAPH free-radical and *D viscosa* infusions at different polyphenol final concentrations. (**B**) Table summarizing the delayed red blood cell half-hemolysis time (h) in the presence of medicinal plant preparations (25 µM) depending on extraction mode. Data are means ± SEM of three independent experiments. ** *p* < 0.01, *** *p* < 0.005 as compared to decoctions.

**Figure 5 antioxidants-09-00959-f005:**
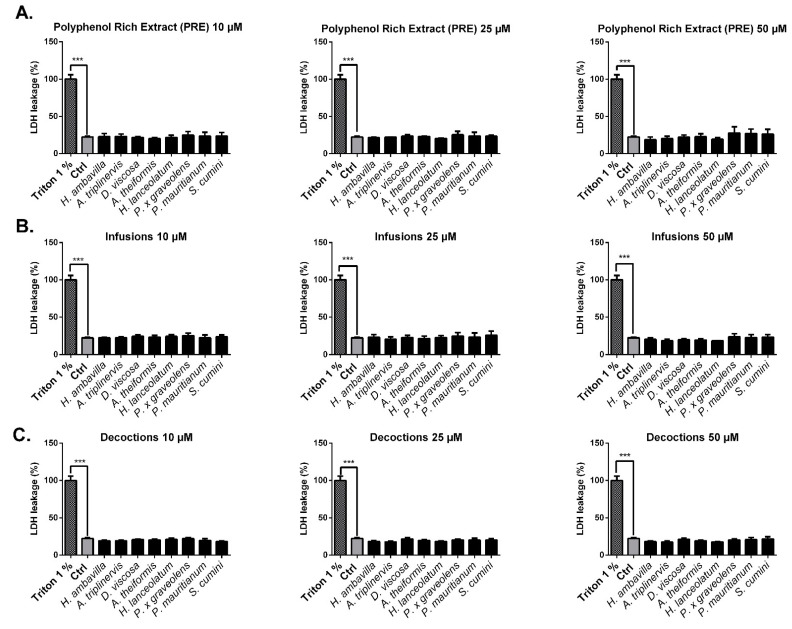
Impact of medicinal plant preparations on 3T3-L1 preadipocytes necrosis. 3T3-L1 preadipocytes were incubated for 24h with polyphenol-rich extracts (PRE) (**A**), infusions (**B**) and decoctions (**C**). LDH release (%) was measured in the supernatant. Triton X100 (1%) was used to induce cell necrosis (considered as 100% necrosis). Absorbance was read at 450 nm. The data shown are means ± SEM of three independent experiments *** *p* < 0.005 as compared to control. LDH: lactate dehydrogenase (an intracellular enzyme whose release into the extracellular medium reflects cell necrosis).

**Figure 6 antioxidants-09-00959-f006:**
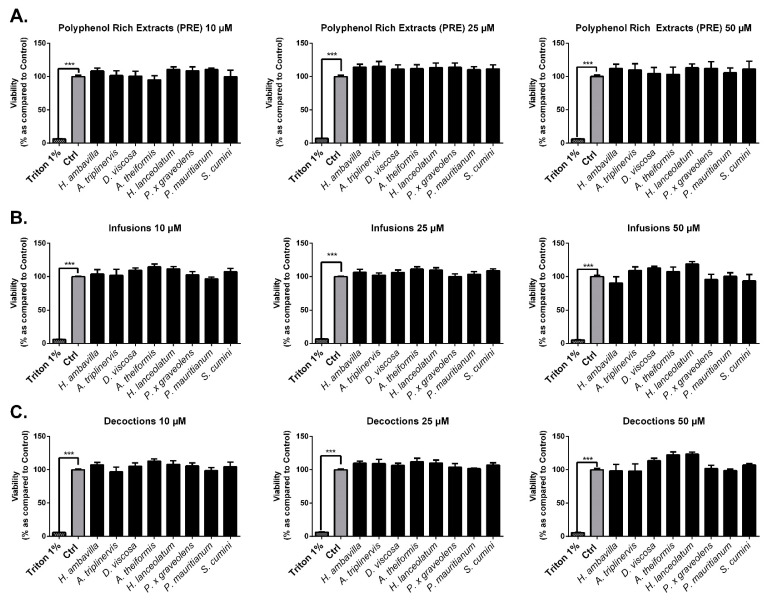
Effects of medicinal plant preparations on 3T3-L1 viability and proliferation. 3T3-L1 preadipocytes were incubated with different medicinal plant extracts: PRE (**A**), infusions (**B**) and decoctions (**C**). Viability (% Control) was assessed by the neutral red method. Triton X100 (1%) was used to induce cell necrosis. Results are presented as means ± SEM of three independent experiments *** *p* < 0.005 as compared to control (Ctrl).

**Figure 7 antioxidants-09-00959-f007:**
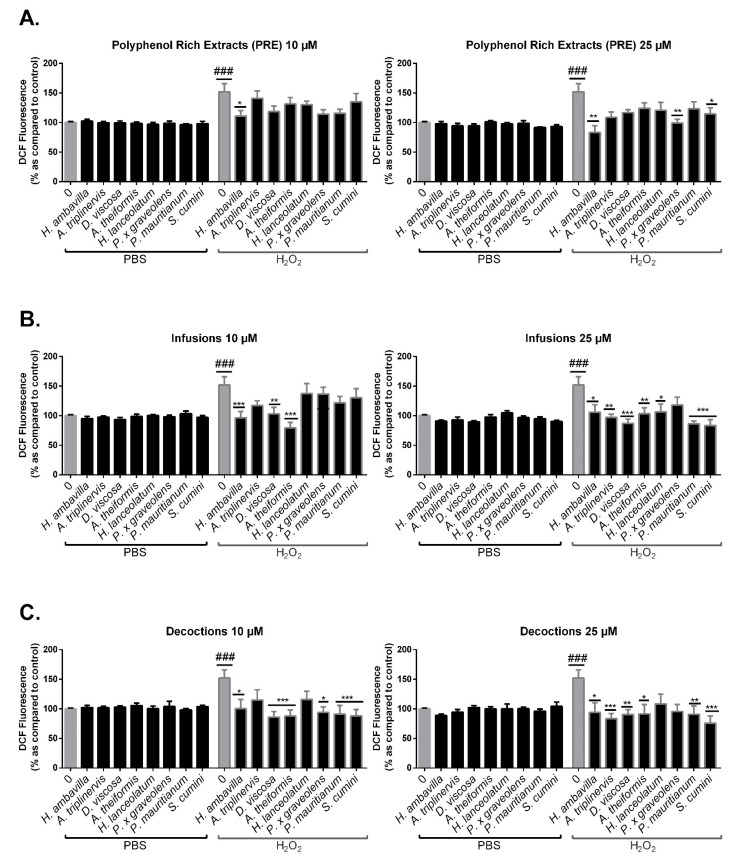
Effects of medicinal plant preparations on intracellular oxidative stress of preadipocytes exposed to H_2_O_2_. 3T3-L1 preadipocytes were incubated for 24 h with PBS, polyphenol-rich extracts (PRE) (**A**), infusions (**B**) and decoctions (**C**) at different concentrations (10–25 µM). DCF fluorescence was measured after 45 min of incubation with 2′,7′-dichlorofluorescein diacetate (DCFH-DA), followed by 1 h of incubation with PBS ± of H_2_O_2_ (200 μM). Data shown are means ± SEM of three independent experiments. ### *p* < 0.005 as compared to PBS alone (left histograms) * *p* < 0.05, ** *p* < 0.01, *** *p* < 0.005 as compared to controls (PBS alone, left histograms or H_2_O_2_, right histograms).

**Table 1 antioxidants-09-00959-t001:** Industrial lots and raw material cropland localization according to the global positioning system.

Medicinal Plant	Industrial Lot	GPS Coordinates
*Ayapana triplinervis*	PAY180111	−21.037151, 55.687405″
*Hubertia ambavilla*	FLAM20171127	−21.131021, 55.640708″
*Psiloxylon mauritianum*	FLBPM 20180427	−20.947334, 55.548979″
*Dodonaea viscosa*	FLBA20171106	−21.059596, 55.507404″
*Aphloia theiformis*	FLCE20171106	−21.059596, 55.507404″
*Hypericum lanceolatum*	FLFJ20171016	−21.140988, 55.641630″
*Syzygium cumini*	JAMBLON230617	−20.540390, 55.26290″
*Pelargonium x graveolens*	FLOGE171113	−21.210391, 55.548644″

**Table 2 antioxidants-09-00959-t002:** UPLC-MS/MS analysis of the polyphenol content of medicinal plant PRE, infusions and decoctions.

Plant	Retention Time (min)	Molecular Weight (Da)	[M-H]^−^/[M-H]^2−^	[M-H]^+^	Assigned Identity	Content in µg/mL			% of Total Composition		
						Extract	Infusion	Decoction	Extract	Infusion	Decoction
*H. ambavilla*	3.4	354	353		Chlorogenic acid isomer	63.6 ± 0.2	0.40 ± 0.3	0.74 ± 0.01	0.51 ± 0.00	0.71 ± 0.01	1.21 ± 0.02
	3.6	370	369		Unidentified	/	/	/	/	/	/
	3.9	354	353		Chlorogenic acid	1337.9 ± 5.5	7.18 ± 0.11	8.02 ± 0.21	10.98 ± 0.04	12.71 ± 0.19	13.06 ± 0.34
	5.0	610	609		Quercetin-hexose-rhamnose	369.8 ± 4.4	0.54 ± 0.01	0.64 ± 0.01	2.95 ± 0.04	0.6 ± 0.02	1.04 ± 0.02
	5.2	464	463		Quercetin-hexose	2473.5 ± 10.5	9.68 ± 0.03	11.43 ± 0.08	19.71 ± 0.08	17.13 ± 0.06	18.61 ± 0.13
	(5.4; 5.7; 5.8)	516	515		Di-caffeoyl quinic acid isomers	4876.5 ± 24.7	23.34 ± 0.15	25.34 ± 0.20	38.85 ± 0.20	41.31 ± 0.27	41.27 ± 0.32
	5.8	682	681		Di-O-caffeoyl-O-[(hydroxy-oxocyclohe-dienyl)acetyl] quinic acid + hydroxy	930.5 ± 13.9	4.34 ± 0.13	17.50 ± 0.32	7.41 ± 0.11	7.68 ± 0.23	6.37 ± 0.26
	(6.3; 6.6)	666	665		Di-O-caffeoyl-O-[(hydroxy-oxocyclohe-dienyl)acetyl] quinic acid isomers	2280.8 ± 2.7	10.13 ± 0.16	10.24 ± 0.03	18.17 ± 0.02	17.93 ± 0.28	16.68 ± 0.06
	7.0	650	649		Di-O-caffeoyl-O-[(hydroxyphenyl)acetyl] quinic acid	178.0 ± 1.1	0.88 ± 0.01	1.08 ± 0.01	1.42 ± 0.01	1.56 ± 0.02	1.76 ± 0.02
*A. triplinervis*	(2.6; 2.9; 3.0; 3.2; 3.5)	372	371		Caffeoyl glucarate isomers	58.6 ± 0.43	1.351 ± 0.008	1.359 ± 0.006	3.47 ± 0.03	12.55 ± 0.08	13.17 ± 0.07
	4.3	534	533		Di-caffeoyl glucarate	69.7 ± 0.6	1.251 ± 0.033	1.312 ± 0.003	4.13 ± 0.04	11.62 ± 0.31	12.72 ± 0.03
	5.0	610	609		Quercetin-hexose-rhamnose or feruoyl hexose	478.8 ± 2.0	2.649 ± 0.009	2.472 ± 0.046	28.37 ± 0.12	24.61 ± 0.08	23.97 ± 0.45
		356	355								
	5.2	464	463		Quercetin hexose	192.9 ± 2.8	1.121 ± 0.015	1.205 ± 0.011	11.43 ± 0.17	10.42 ± 0.14	11.68 ± 0.11
	5.5	550	549		Quercetin-hexose-malonate	232.7 ± 1.7	1.303 ± 0.007	0.997 ± 0.034	13.79 ± 0.10	12.11 ± 0.07	9.67 ± 0.33
	6.0	696	695		Tri-caffeoyl glucarate	212.6 ± 4.4	1.246 ± 0.023	1.278 ± 0.013	12.60 ± 0.26	11.58 ± 0.21	12.39 ± 0.13
	6.4	194	193		Isoferulic acid or thymohydroquinone dimethyl ether	189.05 ± 0.5	0.898 ± 0.018	0.799 ± 0.018	11.20 ± 0.03	8.34 ± 0.17	7.75 ± 0.17
	10.5	/	/	/	Unidentified	/	/	/	/	/	/
	6.9	190		191	Ayapin	31.1 ± 0.4	0.132 ± 0.006	0.122 ± 0.001	1.84 ± 0.02	1.23 ± 0.06	1.18 ± 0.01
	7.5	176		177	Herniarin	222.3 ± 0.7	0.707 ± 0.015	0.671 ± 0.004	13.17 ± 0.04	6.57 ± 0.14	6.51 ± 0.04
*D. viscosa*	(2.6; 3.0; 3.2; 3.5)	372	371		Caffeoyl glucarate isomers	47 ± 0.15	0.759 ± 0.002	1.434 ± 0.003	1.43 ± 0.00	4.54 ± 0.01	4.15 ± 0.02
	3.4	354	353		Chlorogenic acid isomer	47.9 ± 0.6	0.331 ± 0.005	0.480 ± 0.003	1.45 ± 0.02	1.98 ± 0.03	2.66 ± 0.02
	(3.7; 3.8; 4.8)	1152	1151		Procyanidin tetramer type A isomers	646.3 ± 2.47	4.538 ± 0.034	5.002 ± 0.049	19.59 ± 0.08	27.15 ± 0.21	27.73 ± 0.27
	4.0	354	353		Chlorogenic acid	186.2 ± 2.9	0.888 ± 0.017	0.733 ± 0.008	5.64 ± 0.09	5.31 ± 0.10	4.06 ± 0.04
	4.3	578	577		Procyanidin dimer type B	548.5 ± 5.4	2.431 ± 0.035	2.354 ± 0.096	16.63 ± 0.16	14.54 ± 0.21	13.05 ± 0.53
	4.5	290	289		Catechin or epicatechin	388.0 ± 8.7	1.112 ± 0.025	1.118 ± 0.010	11.76 ± 0.26	6.65 ± 0.15	6.20 ± 0.06
	4.5	338	337		Coumaroyl quinic acid	105.3 ± 2.5	0.449 ± 0.009	3.557 ± 0.044	3.19 ± 0.08	2.69 ± 0.05	2.64 ± 0.08
	(4.6; 5.0)	864	863		Procyanidin trimer type A isomers	988.9 ± 6.25	4.389 ± 0.021	5.085 ± 0.05	29.98 ± 0.19	26.25 ± 0.13	28.19 ± 0.27
	5.0	610	609		Quercetin-hexose-rhamnose	59.8 ± 0.9	0.307 ± 0.005	0.329 ± 0.004	1.81 ± 0.03	1.84 ± 0.03	1.82 ± 0.02
	5.3	478	477		Quercetin-glucuronide	30.9 ± 0.5	0.173 ± 0.002	0.197 ± 0.001	0.94 ± 0.02	1.03 ± 0.01	1.09 ± 0.01
	5.4	624	623		Isorhamnetin-hexose-rhamnose	175.4 ± 1.15	0.881 ± 0.010	0.991 ± 0.005	5.86 ± 0.03	5.26 ± 0.06	5.49 ± 0.02
	5.7	492	491		Isorhamnetin-glucuronide	74.3 ± 0.9	0.464 ± 0.007	0.525 ± 0.005	2.25 ± 0.03	2.77 ± 0.04	2.91 ± 0.03
	(8.2; 9.3; 9.5; 9.6; 9.7; 10.7; 10.9)	(416; 330; 454; 484; 398; 436; 466)	(415; 329; 453; 483; 397; 435; 465)		Unidentified	/	/	/	/	/	/
*A. theiformis*	3.5	584	583		Neomangiferin	243.1 ± 0.5	1.56 ± 0.02	1.93 ± 0.030	3.59 ± 0.01	2.76 ± 0.35	2.56 ± 0.04
	3.9	408	407		Iriflophenone-C-hexoside	/	/	/	/	/	/
	4.2	422	421		Mangiferin	6488.6 ± 27.0	54.9 ± 0.10	73.4 ± 0.30	95.71 ± 0.40	97.24 ± 0.18	97.23 ± 0.40
	(4.4; 4.9)	436	435		Homomangiferin or O-methylisomangiferin isomers	47.6 ± 0.85	/	0.16 ± 0.01	0.71 ± 0.01	/	0.21 ± 0.00
	5.2	452	451		Aspalatin	/	/	/	/	/	/
*H. lanceolatum*	3.4	354	353		Chlorogenic acid isomer	598.2 ± 7.0	4.36 ± 0.04	4.11 ± 0.06	3.70 ± 0.04	4.06 ± 0.04	3.71 ± 0.05
	3.9	354	353		Chlorogenic acid	7022.3 ± 11.3	54.54 ± 0.27	53.17 ± 0.21	43.47 ± 0.07	50.78 ± 0.25	47.94 ± 0.19
	4.3	578	577		Procyanidin dimer type B	669.1 ± 21.9	3.56 ± 0.26	4.33 ± 0.22	4.14 ± 0.14	3.31 ± 0.24	3.90 ± 0.20
	4.4	180	179		Caffeic acid	240.5 ± 4.5	1.30 ± 0.02	2.82 ± 0.07	1.49 ± 0.03	1.21 ± 0.02	2.54 ± 0.06
	4.6	866	865		Procyanidin trimer type B	642.3 ± 9.8	3.09 ± 0.29	3.51 ± 0.07	3.98 ± 0.06	2.88 ± 0.27	3.17 ± 0.06
	4.8	1154	1153		Procyanidin tetramer type B	497.8 ± 29.6	2.36 ± 0.10	2.18 ± 0.16	3.08 ± 0.18	2.20 ± 0.09	1.97 ± 0.14
	5.0	610	609		Quercetin-hexose-rhamnose	197.1 ± 1.7	1.16 ± 0.03	1.17 ± 0.04	1.22 ± 0.10	1.08 ± 0.03	1.06 ± 0.04
	(5.2; 5.3)	464	463		Quercetin-hexose isomers	1041.8 ± 2.35	6.13 ± 0.03	6.90 ± 0.04	6.45 ± 0.02	5.71 ± 0.04	6.22 ± 0.05
	5.5	550	549		Quercetin-malonate-hexose	747.2 ± 7.1	3.75 ± 0.04	3.52 ± 0.07	4.62 ± 0.04	3.49 ± 0.04	3.17 ± 0.06
	5.7	448	447		Quercetin-rhamnose	427.8 ± 4.9	2.10 ± 0.03	2.36 ± 0.02	2.65 ± 0.03	1.96 ± 0.03	2.13 ± 0.02
	(5.7; 5.9)	516	515		Di-caffeoyl quinic acid isomers	3440.3 ± 19.4	21.61 ± 0.12	23.00 ± 0.1	21.3 ± 0.24	20.12 ± 0.11	20.74 ± 0.09
	6.0	520	519		Isorhamnetin-acetyl-hexose	38.1 ± 0.5	0.16 ± 0.01	0.21 ± 0.01	0.24 ± 0.00	0.15 ± 0.01	0.19 ± 0.01
	6.2	500	499		Caffoyl coumaroyl quinic acid	252.6 ± 7.3	1.50 ± 0.05	1.70 ± 0.01	1.56 ± 0.05	1.40 ± 0.05	1.53 ± 0.01
	6.4	530	529		Feruoyl caffeoyl quinic acid	153.7 ± 3.5	0.87 ± 0.01	0.99 ± 0.03	0.95 ± 0.02	0.81 ± 0.01	0.89 ± 0.03
	7.1	302	301		Quercetin	187.4 ± 8.7	0.92 ± 0.01	0.93 ± 0.03	1.16 ± 0.05	0.01 ± 0.01	0.84 ± 0.03
*P. x* graveolens	(2.9; 3.5)	372	371		Caffeoyl glucarate isomers	28.2 ± 0.75	0.227 ± 0.004	0.146 ± 0.003	1.73 ± 0.05	2.41 ± 0.04	1.88 ± 0.04
	4.5	612	611		Myricetin derivatives	41.4 ± 1.1	0.198 ± 0.006	0.153 ± 0.005	2.55 ± 0.07	2.10 ± 0.06	1.97 ± 0.06
	4.7	626	625		Myricetin-rhamnose-hexose	55.9 ± 1.1	0.263 ± 0.008	0.220 ± 0.005	3.44 ± 0.07	2.79 ± 0.08	2.83 ± 0.06
	4.8	480	479		Myricetin-hexose	103.6 ± 0.9	0.520 ± 0.005	0.425 ± 0.011	6.37 ± 0.06	5.51 ± 0.05	5.47 ± 0.14
	4.9	596	595		Quercetin-pentose-hexose	268.3 ± 2.1	1.574 ± 0.019	1.312 ± 0.010	16.51 ± 0.13	16.69 ± 0.20	16.87 ± 0.13
	5.1	610	609		Quercetin-hexose-rhamnose	122.3 ± 2.2	0.761 ± 0.008	0.636 ± 0.002	7.52 ± 0.14	8.07 ± 0.08	8.18 ± 0.03
	5.1	450	449		Myricetin-pentose	43.4 ± 1.3	0.184 ± 0.009	0.150 ± 0.002	2.67 ± 0.08	1.95 ± 0.10	1.93 ± 0.03
	5.2	464	463		Myricetin-rhamnose	91.1 ± 0.7	0.456 ± 0.009	0.381 ± 0.004	5.60 ± 0.04	4.84 ± 0.10	4.90 ± 0.05
	(5.2; 5.3)	464	463		Quercetin-hexose isomers	513.6 ± 2.9	3.058 ± 0.006	2.558 ± 0.009	31.6 ± 0.18	32.43 ± 0.06	32.9 ± 0.12
	5.4	594	593		Kaempferol-hexose-rhamnose	16.4 ± 0.4	0.089 ± 0.002	0.067 ± 0.005	1.01 ± 0.02	0.94 ± 0.02	0.86 ± 0.06
	(5.5; 5.6)	448	447		Kaempferol-hexose isomers	112.4 ± 0.8	0.655 ± 0.004	0.534 ± 0.004	5.07 ± 0.00	6.95 ± 0.04	6.87 ± 0.05
	5.5	434	433		Quercetin pentose	189.2 ± 2.9	1.092 ± 0.010	0.877 ± 0.004	11.64 ± 0.18	11.58 ± 0.11	11.28 ± 0.05
	5.8	418	417		Kaempferol-pentose	44.6 ± 0.6	0.252 ± 0.003	0.201 ± 0.003	2.74 ± 0.04	2.67 ± 0.03	2.58 ± 0.04
	6.2	318	317		Myricetin	/	/	/	/	/	/
	7.1	302	301		Quercetin	19.3 ± 0.4	0.090 ± 0.002	0.101 ± 0.002	1.19 ± 0.02	0.95 ± 0.02	1.30 ± 0.03
	7.7	286	285		Kaempferol	5.8 ± 0.2	0.011 ± 0.001	0.015 ± 0.001	0.36 ± 0.01	0.12 ± 0.01	0.19 ± 0.01
*S. cumini*	2.4	170	169		Gallic acid	210.7 ± 6.8	1.931 ± 0.039	6.351 ± 0.084	19.54 ± 0.63	21.03 ± 0.42	27.82 ± 0.37
	(2.5; 2.9)	634	633		HHDP-galloyl-hexose isomers	59.5 ± 0.8	1.025 ± 0.016	2.969 ± 0.021	5.52 ± 0.07	11.16 ± 0.18	13.01 ± 0.18
	2.6	484	483		Di-galloyl-hexose	7.7 ± 0.3	0.090 ± 0.002	0.143 ± 0.003	0.71 ± 0.03	0.98 ± 0.02	0.63 ± 0.01
	3.0	934	933/466		Vescalagin/di-galloyl-hexose	30.8 ± 0.7	0.255 ± 0.014	0.533 ± 0.009	2.86 ± 0.06	2.78 ± 0.15	2.33 ± 0.04
		484	483								
	(3.2; 3.7; 4.2; 4.5)	784	783		Bis-HHDP-hexose isomers	63.1 ± 0.7	0.720 ± 0.006	1.253 ± 0.015	5.85 ± 0.06	3.14 ± 0.02	1.41 ± 0.03
	(3.2; 3.6)	802	801		Galloyl tannin isomers	53.1 ± 0.9	0.775 ± 0.010	2.583 ± 0.023	4.92 ± 0.08	8.44 ± 0.11	11.32 ± 0.11
	3.4	934	933/466		Castalagin/ellagitannin	50.7 ± 0.7	0.623 ± 0.007	0.968 ± 0.029	4.70 ± 0.06	6.79 ± 0.08	4.24 ± 0.013
		1418	1417/708								
	3.9	952	951		Trisgalloyl-HHDP-hexose/ellagitannin	53.7 ± 1.7	0.420 ± 0.009	0.662 ± 0.027	4.98 ± 0.16	4.57 ± 0.10	2.90 ± 0.12
		1418	1417/708								
	4.0	1086	1085		Digalloyl-Gallagyl-hexose	97.2 ± 0.85	0.802 ± 0.008	1.511 ± 0.013	9.01 ± 0.08	5.29 ± 0.13	4.18 ± 0.07
	4.3	952	951		Trisgalloyl-HHDP-hexose	51.7 ± 2.3	0.370 ± 0.017	0.814 ± 0.035	4.79 ± 0.21	4.03 ± 0.19	3.57 ± 0.15
	(4.8; 5.0)	434	433		Ellagic acid -pentose isomers	209.1 ± 2.8	1.086 ± 0.017	2.039 ± 0.033	19.39 ± 0.26	4.77 ± 0.10	3.86 ± 0.19
	5.2	302	301		Ellagic acid	191.0 ± 3.5	1.083 ± 0.038	3.005 ± 0.072	17.71 ± 0.32	11.79 ± 0.41	13.16 ± 0.32
*P. mauritianum*	5.3	464	463		Quercetin-hexose	166.4 ± 6.6	0.796 ± 0.026	1.235 ± 0.026	21.74 ± 0.86	25.43 ± 0.83	32.85 ± 0.69
	5.7	448	447		Kaempferol-hexose	274.9 ± 10.6	1.094 ± 0.035	1.208 ± 0.053	35.91 ± 1.38	34.95 ± 1.12	32.13 ± 1.41
	6.1	506	505		Quercetin-acetyl- hexose	66.7 ± 3.9	0.256 ± 0.003	0.283 ± 0.015	8.71 ± 0.51	8.18 ± 0.10	7.53 ± 0.40
	6.4	490	489		Kaempferol-acetyl-hexose	165.3 ± 7.9	0.634 ± 0.012	0.662 ± 0.002	21.59 ± 1.03	20.26 ± 0.38	17.61 ± 0.59
	6.9	460	459		Kaempferol-acetyl-pentose	13.2 ± 0.4	0.039 ± 0.002	0.042 ± 0.002	1.72 ± 0.05	1.25 ± 0.06	1.12 ± 0.05
	7.0	474	473		Kaempferol-acetyl-rhamnose	49.6 ± 1.0	0.219 ± 0.005	0.230 ± 0.002	6.48 ± 0.13	7 ± 0.16	6.12 ± 0.06
	7.7	286	285		Kaempferol	29.4 ± 1.5	0.092 ± 0.006	0.100 ± 0.005	3.84 ± 0.20	2.94 ± 0.19	2.66 ± 0.13
	8.8	488	487		Asiatic acid	/	/	/	/	/	/
	(9.2; 10.3)	(238; 266)	(237; 265)		Unidentified	/	/	/	/	/	/

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
