# Peer review of "Evaluation of Polyphenol Content and Antioxidant Capacity of Aqueous Extracts from Eight Medicinal Plants from Reunion Island: Protection against Oxidative Stress in Red Blood Cells and Preadipocytes"

_antioxidants, 2020, doi:10.3390/antiox9100959_

Round 1
Reviewer 1 Report
The article concerns the studying of polyphenol content and antioxidant capacity of eight medicinal plants from Reunion Island. The content was determined by a color reaction and Ultra Performance Liquid Chromatography coupled to LSI-mass spectrometry. The antioxidant capacity was found by the measuring DPPH reduction and the studying of protective effects of different extracts on red blood cells or preadipocytes exposed to oxidative stress. The UPLC/LSI MS analysis showed the presence of so typical bioactive polyphenols such as quercetin, chlorogenic acid, procyanidin and mangiferin. The antioxidant capacity of the studied extracts demonstrated a dose-dependent effect whatever the extraction procedure. The article seems to be interesting and appropriate for publishing but has several imperfections.
- The authors can not write Latin names of the plants correctly. The point should be placed just after shortage of the genus name. For example, S. cumini instead of S cumini. Such errors are presented at the lines 206 (Fig. 1), 214, 220, 232 (Table 2), 233, 234, 238, 239, 242, 245, 246, 248, 250, 254, 255, 276, 287 (Fig. 2), 295, 298, 303, 304, 305 (Fig. 3), 312, 313, 319, 340 (Fig. 4), 353, 359, 384 (Fig. 5), 402, 40. Moreover the Latine name at the Lines 227–229 should be written in italic.
- Table 2 is not accurate. A lot of disrupted words.
- Line 176. Replace “μl” with “μL”.
- Fig. 1 is badly organized. The order of species at the diagrama A and B should be the same in order to provide a possibility to compare the data for different kinds of extracts.
- Line 226. Replace “Analysis” with “analysis”.
- Line 131. Replace “2.1-100 mm2” with “2.1 × 100 mm”
The results in the Table 2 are badly organized. They should be recalculated and the data in % from total sum should be presented for comparison of the data for different extract. It is especially important because “the bioavailability of polyphenols contained in PRE and water-based extractions is different” (Lines 396, 397).
- The most serious flaw of the article is absolutely wrong system of literature citation and formatting of the references. Just download any fresh article from the Antioxidants and reorganize the citation system and references strictly along with the journal style.
General conclusion: Major revision.
Author Response
Reviewer #1
The article concerns the studying of polyphenol content and antioxidant capacity of eight medicinal plants from Reunion Island. The content was determined by a color reaction and Ultra Performance Liquid Chromatography coupled to LSI-mass spectrometry. The antioxidant capacity was found by the measuring DPPH reduction and the studying of protective effects of different extracts on red blood cells or preadipocytes exposed to oxidative stress. The UPLC/LSI MS analysis showed the presence of so typical bioactive polyphenols such as quercetin, chlorogenic acid, procyanidin and mangiferin. The antioxidant capacity of the studied extracts demonstrated a dose-dependent effect whatever the extraction procedure. The article seems to be interesting and appropriate for publishing but has several imperfections.
We thank the Reviewer #1 for her/his suggestions. We have answered point by point to each comment/suggestion.
- The authors can not write Latin names of the plants correctly. The point should be placed just after shortage of the genus name. For example, cuminiinstead of S cumini. Such errors are presented at the lines 206 (Fig. 1), 214, 220, 232 (Table 2), 233, 234, 238, 239, 242, 245, 246, 248, 250, 254, 255, 276, 287 (Fig. 2), 295, 298, 303, 304, 305 (Fig. 3), 312, 313, 319, 340 (Fig. 4), 353, 359, 384 (Fig. 5), 402, 40. Moreover the Latine name at the Lines 227–229 should be written in italic.
The modification of the Latin names of the plants was carried out according to the recommendations of the reviewer. These changes have been made in figures and tables and have been highlighted throughout the article.
- 1 is badly organized. The order of species at the diagrama A and B should be the same in order to provide a possibility to compare the data for different kinds of extracts.
In order to facilitate the comparison between polyphenol contents, the order of species has been modified according to the reviewer's recommendations.
- Table 2 is not accurate.
- A lot of disrupted words. The results in the Table 2 are badly organized. They should be recalculated and the data in % from total sum should be presented for comparison of the data for different extract. It is especially important because “the bioavailability of polyphenols contained in PRE and water-based extractions is different” (Lines 396, 397).
The format of Table 2 has been modified. The reviewer's remark was taken into account and the results were expressed as percentage of the total sum. This new result organization is much more relevant and enables to clearly identify the main polyphenols present in the different plant preparations.
- Line 176. Replace “μl” with “μL”.
- Line 226. Replace “Analysis” with “analysis”.
- Line 131. Replace “2.1-100 mm2” with “2.1 × 100 mm”
The errors related to symbol formatting and typography cited by the reviewer have been corrected and highlighted in the manuscript.
- The most serious flaw of the article is absolutely wrong system of literature citation and formatting of the references. Just download any fresh article from the Antioxidants and reorganize the citation system and references strictly along with the journal style.
An article recently published in the Journal Antioxidants was used to properly edit the bibliography according to the requirements of the Journal.
General conclusion: Major revision.
Reviewer 2 Report
Antioxidant capacity depends on the type of compound.
And the extract that contains the compound with high antioxidant power in high concentration has high antioxidant power.
To ensure reproducibility of studies, antioxidant capacity needs to be examined at the compound level.
When discussing the antioxidant capacity of extracts, it is necessary to clarify the meaning of being an extract (mixture).
The novelty of these plants themselves is retained, but not the novelty of their
constituents.
Author Response
Reviewer #2
Antioxidant capacity depends on the type of compound.
And the extract that contains the compound with high antioxidant power in high concentration has high antioxidant power.
To ensure reproducibility of studies, antioxidant capacity needs to be examined at the compound level.
When discussing the antioxidant capacity of extracts, it is necessary to clarify the meaning of being an extract (mixture).
The novelty of these plants themselves is retained, but not the novelty of their
constituents.
We thank the Reviewer #2 for her/his comments. We have modified our manuscript accordingly.
The purpose of our study was to determine whether 8 traditional medicinal plants contain bioactive compounds. Thank you for acknowledging that the novelty of our study relies on the nature of the plants analysed. These plants are listed in the French Pharmacopoeia, which allows their use for phytotherapeutic purposes. Our work is part of the contribution of new data on rare medicinal plants, with promising health benefits, which are widely consumed by the population.For example, we describe for the first time the composition and activity of several plants such as P. mauritianum, A. theiformis and H. lanceolatum that are almost unknown in the literature.
We agree that medicinal plant preparations (PREs, infusions and decoctions) constitute a mixture whose biological effects involve synergistic mechanisms between compounds. This is a limitation of our study, which is primarily intended to support the first basic knowledge on the antioxidant properties of traditionally used medicinal plants.
This limitation has been added in the new version of the manuscript. " One limitation of our study is that we analyzed plant extracts that implies a mixture of several antioxidant compounds at different concentrations acting alone or in synergy. Further investigations would be needed to investigate the individual contribution of each component to the global antioxidant activity."
Reviewer 3 Report
The paper by Checkouri and colleagues focuses on the antioxidant properties of 8 medicinal plants from Reunion Island. The topic is very interesting and the article is well written and nice to read. I think that the paper could be accepted after minor revisions. I report my comments below.
Introduction is well written and plant selection criteria are clearly explained. Some terms may sound a bit outdated (e.g. I would suggest substituting “numerous” with “several”). Check the symbol of radical. In my opinion, references should be updated. For example, when talking about oxidative stress in general terms or bioactive natural compounds, I would suggest citing papers belonging to the last decade. See for example:
- Sies, H. et al. Reactive oxygen species (ROS) as pleiotropic physiological signalling agents. Nat. Rev. Mol. Cell Biol. 2020, 21, 363–383
- Zanforlin, E. et al. An overview of new possible treatments of alzheimer’s disease, based on natural products and semi‐synthetic compounds. Curr. Med. Chem. 2017, 24, 3749–3773
Materials and methods section is sufficiently detailed and well organized. I have a concern regarding the sentence “The polyphenol-rich supernatant (ranging from 7 to 15 mL)”: the authors should briefly state the reasons of this volume variability.
In Results and discussion, results are clearly discussed. Concerning Figure 2, legend is too big with respect to the size of the figure. I would suggest to comment more on the expected molecular mechanism (radical scavenging, etc.) for the identified and quantified natural compounds. In this connection, with respect to paragraph 7.3.2, authors should comment that peroxide inactivation is one of the relevant antioxidant mechanisms underlying the activity of natural and semi-synthetic compounds. See for example:
- Pavan, V. et al. Antiproliferative activity of Juglone derivatives on rat glioma, Nat Prod Res. 2016, http://dx.doi.org/10.1080/14786419.2016.1214830
In Conclusions, I think that the final sentence is quite “strong” (“These results scientifically support…”): I suggest rephrasing.
Concerning the supplementary file, I think that this additional file is not necessary: the information reported there could be simply included in the main manuscript, with the corresponding references.
Author Response
Reviewer #3
The paper by Checkouri and colleagues focuses on the antioxidant properties of 8 medicinal plants from Reunion Island. The topic is very interesting and the article is well written and nice to read. I think that the paper could be accepted after minor revisions. I report my comments below.
We thank the Reviewer #3 for her/his suggestions. We have answered point by point to each comment/suggestion.
- Introduction is well written and plant selection criteria are clearly explained. Some terms may sound a bit outdated (e.g. I would suggest substituting “numerous” with “several”). Check the symbol of radical.
The word "numerous" has been replaced by "several" as requested by the reviewer. These changes have been highlighted in the article. A modification was made for the free radical symbols according to the article of Shirley et al. (“Oxidative Stress and the Use of Antioxidants in Stroke”) published in Antioxidants in 2014. The modification has been highlighted in the article.
In my opinion, references should be updated. For example, when talking about oxidative stress in general terms or bioactive natural compounds, I would suggest citing papers belonging to the last decade. See for example:
- Sies, H. et al. Reactive oxygen species (ROS) as pleiotropic physiological signalling agents. Rev. Mol. Cell Biol. 2020, 21, 363–383
- Zanforlin, E. et al. An overview of new possible treatments of alzheimer’s disease, based on natural products and semi‐synthetic compounds. Med. Chem. 2017, 24, 3749–3773
Some references have been updated as requested by the reviewer. In particular, review articles by Sies et al. (2017 and 2020) have been added to the paragraph related to oxidative stress. Additionally, recent studies were added to the bibliography: first the review of Zanforlin et al. 2017, but also the review of Arya et al., 2020 to illustrate the effects of polyphenols on inflammatory pathways (mostly NFkB and TLRs signaling). The review from D’Angelo et al. 2020 was also added because in addition to addressing polyphenols anti-inflammatory mechanisms, this study also highlights the antioxidant mechanisms of redox enzyme activity and production. These modifications allowed us to add more recent and more complete data to the bibliography of our manuscript.
- Materials and methods section is sufficiently detailed and well organized. I have a concern regarding the sentence “The polyphenol-rich supernatant (ranging from 7 to 15 mL)”: the authors should briefly state the reasons of this volume variability.
During the extraction of the compounds, some plant material proved to be more hydrophobic than others (for example, this was the case for P. x graveolens). Therefore, the volume of solution recovered depended on the plant studied. During the three independent experiments, the volumes recovered were reproducible and practically identical for a given plant. In order to explain the variability between volumes we added "depending on the level of solvent retention by the plant material" in our manuscript.
- In Results and discussion, results are clearly discussed. Concerning Figure 2, legend is too big with respect to the size of the figure.
The legend of the figure 2 was modified so that characters and symbols are clearly identifiable.
- I would suggest to comment more on the expected molecular mechanism (radical scavenging, etc.) for the identified and quantified natural compounds. In this connection, with respect to paragraph 7.3.2, authors should comment that peroxide inactivation is one of the relevant antioxidant mechanisms underlying the activity of natural and semi-synthetic compounds. See for example:
Pavan, V. et al. Antiproliferative activity of Juglone derivatives on rat glioma, Nat Prod Res. 2016, http://dx.doi.org/10.1080/14786419.2016.1214830
To comment the figure 2, the following paragraph was added to support more information on antioxidant activity “This antioxidant effect is the result of the radical scavenging by polyphenol reducing extremities (hydroxyl groups). As described by Villaño et al., polyphenols can exert antioxidant activity by direct abstraction of phenol H-atom with the nitrogen of the DPPH or by undergoing an electron transfer to the free radical. The number of -OH group available is also an important factor that impacts the antioxidant activity. [46]”
In addition to the interpretation of figure 5 we added the paragraph “ These results point out the importance of the antioxidant activity of medicinal plant bioactive compounds that are known to exert peroxide inactivation leading to a protective effect on oxidative stress damage (mitochondrial stress, activation of redox signaling, cell death) [60,62]”. The reference from Pavan et al. (2016) suggested by the reviewer was added with another reference from Sies et al., (2017) that specifies the cellular mechanisms resulting from oxidative stress conditions.
In Conclusions, I think that the final sentence is quite “strong” (“These results scientifically support…”): I suggest rephrasing.
The final sentence of the conclusion has been rephrased as follows: " These results scientifically support the potential of the traditional use of Reunion Island medicinal plant preparations (infusions or decoctions) for limiting oxidative stress, involved in metabolic disorders and associated complications."
Concerning the supplementary file, I think that this additional file is not necessary: the information reported there could be simply included in the main manuscript, with the corresponding references.
We agree that this additional file is important. If the editor allows us to add more reference in the manuscript, this information will be added in the main document.
Round 2
Reviewer 1 Report
The authors have corrected the manuscript stricly along with my notes. Only one minor correction is necessary:
Lines 213, 219 and 271. The "and" should not be in italic. Normalized them please.
The manuscript should be published after very minor correction without next tour of the reviewing.
Author Response
We thank the Reviewer #1 for her/his comments. We have modified our manuscript accordingly.
The word “and” written in italic was normalized at lines 213, 219 and 271 according to the reviewer comments. The modifications were highlighted.
Reviewer 2 Report
The effect of multi-component extracts is difficult to guarantee the reproducibility of the study.
The effects of compounds need to be considered from the perspective of ensuring reproducibility of the study.
Author Response
The traditional use of plant extract such as herbal tea is based on their multi-component composition. The aim of our study is to investigate the composition and potential biological function of these extracts in the conditions used by the population. We agree that it would be of interest to investigate each compound individually but this was not the goal of our study.
We also agree that plant metabolite production depends on environmental conditions and thus that plant extract composition and bioactivity may vary according to these factors. However, our study is based on industrial lots of plants, cultivated on areas with similar environmental conditions and follows the same industrial process in order to optimize the reproducibility.
In another study of our group by Taile et al (https://doi.org/10.3390/antiox9070573) published in the journal Antioxidants on A. triplinervis and D. viscosa from a different terroir showed similar results in terms of composition:
- triplinervis: protocatechuic acid, caffeic acid, feruoyl-hexoside, quercetin hexoside, tricaffeoyl-glucarate, ayapin, ayapanin, isoferulic acid
- viscosa: Procyanidin tetramer type A, procyanidin dimer B2, procyanidin trimer type A, isorhamnetin-hexoside-rhamnoside
The polyphenol contents for D. viscosa are similar (3g GAE/ 100g plant) despite the difference in the cultivation/ drying/milling and extraction process of the plants.
Concerning A. triplinervis, the contents described in Taile et al. article are 0.68 g / 100 g plant against 1 g / 100 g plant in our article. The difference may come from the industrial processes used to optimize the conservation of polyphenols during processing into the final product. Another reason for this difference could also be the cultivation conditions.
This limitation has been added in the new version of the manuscript. "In addition, depending on environmental conditions, industrial process (drying, milling, conditioning) and extraction method, the composition of plant herbal tea may slightly vary from a preparation to another but the main components and biological function should be globally similar."